# Assessment of the Use of Multi-Channel Organic Electrodes to Record ENG on Small Nerves: Application to Phrenic Nerve Burst Detection

**DOI:** 10.3390/s21165594

**Published:** 2021-08-19

**Authors:** Yvan Avdeew, Victor Bergé-Laval, Virginie Le Rolle, Gabriel Dieuset, David Moreau, Loïg Kergoat, Benoît Martin, Christophe Bernard, Christian Gestreau, Alfredo Hernández

**Affiliations:** 1Univ Rennes, Inserm, LTSI-UMR 1099, F-35000 Rennes, France; yvan.avdeew@univ-rennes1.fr (Y.A.); gabriel.dieuset@univ-rennes1.fr (G.D.); benoit.martin@univ-rennes1.fr (B.M.); alfredo.hernandez@univ-rennes1.fr (A.H.); 2Institut de Neurosciences des Systèmes (INS), INSERM, UMR-1106, Aix-Marseille Université Marseille, F-13007 Marseille, France; victor.berge-laval@etu.univ-amu.fr (V.B.-L.); kergoatloig@gmail.com (L.K.); christophe.bernard@univ-amu.fr (C.B.); christian.gestreau@univ-amu.fr (C.G.); 3Mines Saint-Etienne, Centre CMP, Département BEL, F-13541 Gardanne, France; david.moreau@emse.fr

**Keywords:** neuromodulation, electroneurogram, organic electrode, suction electrode, phrenic nerve

## Abstract

Effective closed-loop neuromodulation relies on the acquisition of appropriate physiological control variables and the delivery of an appropriate stimulation signal. In particular, electroneurogram (ENG) data acquired from a set of electrodes applied at the surface of the nerve may be used as a potential control variable in this field. Improved electrode technologies and data processing methods are clearly needed in this context. In this work, we evaluated a new electrode technology based on multichannel organic electrodes (OE) and applied a signal processing chain in order to detect respiratory-related bursts from the phrenic nerve. Phrenic ENG (pENG) were acquired from nine Long Evans rats in situ preparations. For each preparation, a 16-channel OE was applied around the phrenic nerve’s surface and a suction electrode was applied to the cut end of the same nerve. The former electrode provided input multivariate pENG signals while the latter electrode provided the gold standard for data analysis. Correlations between OE signals and that from the gold standard were estimated. Signal to noise ratio (SNR) and ROC curves were built to quantify phrenic bursts detection performance. Correlation score showed the ability of the OE to record high-quality pENG. Our methods allowed good phrenic bursts detection. However, we failed to demonstrate a spatial selectivity from the multiple pENG recorded with our OE matrix. Altogether, our results suggest that highly flexible and biocompatible multi-channel electrode may represent an interesting alternative to metallic cuff electrodes to perform nerve bursts detection and/or closed-loop neuromodulation.

## 1. Introduction

Neuromodulation is a therapeutic approach used in a number of pathologies including neurodegenerative and neuropsychiatric disorders [1,2,3], epilepsy [3,4], neural prostheses [5,6], chronic pain [7], etc. Regarding all of these clinical applications, a common difficulty is to provide effective therapy while minimizing side effects. In fact, delivering an optimal neuromodulation can be particularly complex, because the evoked effect highly depends on (i) the delivered stimulation parameters (current amplitude, frequency, etc.) [8], (ii) the individual response of a given patient to these parameters [8], and (iii) the used electrode configurations and technologies [9]. It has been shown that closed-loop control may be a promising approach to optimize the response to the therapy in a subject-specific, adaptive manner in order to minimize side effects [10]. Closed-loop neuromodulation therapy requires the definition of appropriate control variables and the electroneurogram (ENG), acquired from implanted electrodes, may be a pertinent source of information to this goal [11]. However, improved electrode technologies and data processing methods are needed for the acquisition of this ENG and for deriving useful control variables.

Two types of electrodes are used for peripheral nerve stimulation and/or recording, extraneural electrodes in contact with the epineurium and intraneurals also called intrafascicular electrodes [12,13,14]. Extraneural electrodes are more commonly used in clinical practice [13,15]. Electrode designs and implanted materials are highly variable depending on clinical needs. Different methods of implantation and nerve cuffing can be used for extraneural electrodes, with the most commonly used being metallic spiral cuff electrodes and split-cylinder cuff electrodes [16]. Indeed, such metallic spiral nerve cuff electrodes have been used to record ENG data from small nerves, such as the phrenic nerve [17]. However, the design, materials, and implantation methods of those electrodes need to be improved, particularly regarding their adjustment to the morphology of the nerve, rigidity, difficulty to install, lose contact with the nerve, and/or nerve compression damages [18,19,20,21]. There is also a need for improving spatial resolution of peripheral nerve interfaces [22]. There are some reports of selective stimulation using large metallic multicontact cuff electrodes, such as the one adapted to the dog hypoglossal nerve trunk [23].

Moreover, conjugated polymers provide many benefits over metal electrodes in terms of compliance and biocompatibility as well as selectivity and sensitivity. Doped with polystyrene sulfonate (PSS), PEDOT:PSS has a greatly enhanced conductivity, chemical stability, and biocompatibility [24]. PEDOT:PSS was shown to be a prime material for polymer-based bio-electronic devices as its application on various cultured cell lines proved to preserve their survivability [25], and it was also used to both record and stimulate cortical areas [24,26] as well as peripheral nerves [27,28]. The substrate material of this electrode is parylen C, a polymer known for his flexibility [29,30], and his biocompatibility in both acute and chronic implantation as well as his minimal inflammatory reaction compared to PDMS, another biocompatible polymer used in chronic implantable electrodes [31].

To our knowledge, multicontact OE have not been used to record from small nerves, although this technology may have several advantages over metal electrodes, as discussed. In this paper, we choose to record from the phrenic nerve (PHR) which plays a crucial role in breathing. The objectives of our study were (1) to record pENG signals using a 16 channels custom-design OE adapted to the PHR nerve, (2) to analyze PHR burst detection in a multichannel basis, and (3) to analyze the performance of the OE in comparison with a suction electrode (SE) used as a gold standard.

This paper is organized as follows: Section 2 details the description of the experimental protocol and methods for data processing and comparison. Results on all datasets and a representative example are presented in Section 3. Discussion on the main findings and limitations of the study are given in Section 4.

## 2. Methods

### 2.1. In Situ Preparation

All experiments were performed in Long Evans rats (*n* = 9, P17-25, body weight 50–110 g) of either sex, using the arterially perfused working heart-brainstem or in situ preparation, as described previously [32,33]. This preparation allows the brainstem to be well oxygenated, maintaining a normal blood pH and resulting in a physiological eupneic pattern of respiratory motor activity [34]. Briefly, rats were deeply anesthetized with isoflurane (1-chloro-2,2,2-trifluoroethyl-difluoromethylether; Baxter). Each animal was transected below the diaphragm and decerebrated at the precollicular level once respiration was suppressed and the animal failed to respond to noxious pinch to the tail or toe. After transfer to an ice-cooled area (5 ${}^{\circ }$C) and equilibrated with 95% O${}_{2}$ and 5% CO${}_{2}$ aCSF (composition below), the skin and the lungs were removed. The cerebellum was also removed. The left PHR was prepared for recording (see Figure 1). Following these initial procedures, the preparation was transferred to a custom made recording chamber. The descending aorta was cannulated and perfused with warmed and carbogen-gassed aCSF (33 ${}^{\circ }$C) using a peristaltic pump (Watson-Marlow) via a double-lumen catheter connected to the aortic perfusion cannula and a blood pressure monitor (WPI, USA) at a flow rate of 24–32 mL/min. The aCSF is composed of (in mM) 125 NaCl, 3 KCl, 1.25 KH${}_{2}$PO${}_{4}$, 2.5 CaCl${}_{2}$, 1.25 MgSO${}_{4}$, 25 NaHCO${}_{3}$, and 10 D-glucose (1.25% Ficoll), maintained at 7.35 pH by gassing with a 90% O${}_{2}$ and 5% CO${}_{2}$ carbogen. Filtered and passed through bubble traps, the perfusate was re-collected after leaking from the preparation and re-circulated after its re-oxygenation. Cardiac activity resumed within seconds and rhythmic contractions of respiratory muscles returned 2 to 5 min after the start of perfusion. Respiratory-related movements were abolished by using 250 μL of vecuronium bromide (3–30 μg/mL; Organon) dissolved in aCSF to avoid muscular activity-related artifacts, but it does not affect the central activity of the respiratory network recorded on the nerves. The perfusion flow was then adjusted to obtain an identifiable three-phase respiratory pattern, which was assessed by the recording of the phrenic, vagus and hypoglossal nerve (not shown), innervating the diaphragm, the larynx and the tongue respectively, attesting of the state of the respiratory network (i.e., Inspiration, early-expiration and late-expiration).

### 2.2. Organic Electrode for ENG Recording

The recording grid is composed of 16 (4 × 4) OE with a size of 20 μm × 20 μm. Each OE will be noted here as $C_{i,j}$ (row *i* column *j*, as shown in Figure 2). The recording sites of the OE are disposed in 200 μm × 100 μm rectangles. The OE was connected to the recording instrument with a zero insertion force (ZIF) connector with a 250 μm pitch. The device has a strip at the end and a hole of 600 μm × 550 μm about 2 mm away from the center of the grid. The grid is rolled around the PHR and locked by sliding the strip inside the hole. OE grid fabrication process was described previously [35,36]. A 3 μm thick parylene (PaC) layer (SCS Labcoater 2) is deposited on top of a clean glass slide. Then, metallic electrodes, interconnects, and contact pads are patterned by photolithography (SUSS MBJ4 Contact Aligner). First, AZnLOF2070 negative photoresist is patterned on top of the PaC. Then, a 10 nm thick chromium layer (for a better gold adhesion) and a 100 nm thick gold layer are deposited by thermal evaporation (Suss Microtec, MJB4 Mask Aligner). Finally, the metallic pattern is obtained by lifting off the resist in aceton. A second 3 μm PaC is deposited to act as insulator. Then, the outline of the device is patterned by photolithography using the positive photoresist AZ9260, followed by reactive ion etching of both PaC layers with an oxygen plasma (Oxford, Plasmalab 80 Plus). A sacrificial 3 μm thick PaC layer is deposited and openings in the PaC are patterned, on top of the electrodes and the contact pads, by photolithography using AZ9260 then etched by reactive ion etching with the oxygen plasma. A mixture of PEDOT:PSS (Heraeus Clevions PH1000), ethylene glycol, dodecylbenzene sulfonic acid, and 3-glycidoxypropyltrimethoxysilane is then deposited by spin-coating and baked at 110 ${}^{\circ }$C for 1 min. The sacrificial layer is peeled off, leaving PEDOT:PSS only on top of the OEs and the contact pads. Then, the PEDOT:PSS is hard baked at 140 ${}^{\circ }$C for 1 h. Finally, the device is put in water and delaminates from the glass slide, to obtain the flexible device.

### 2.3. Electrochemical Impedance Spectroscopy

Electrochemical impedance spectroscopy (EIS) measurements were performed in a PBS solution, with an Autolab PGSTAT128N. PEDOT:PSS-coated electrodes were used as the working electrodes and an Ag/AgCl electrode as the counter electrode. A 10 mV sinusoidal voltage was applied at a frequency ranging from 10 to 100,000 Hz (Figure 3).

### 2.4. Nerve Recordings

In each experiment, the left PHR, and the right vagus and hypoglossal nerves were dissected and cut to record the respiratory motor activity using glass suction electrodes (gold standard). SE was made from borosilicate glass capillaries tubes (OD 1.5 mm, WPI), pulled with a vertical pipette puller (Model 720, KOPF). The tip of the electrode was sectioned at a diameter fitting the recorded nerve (200 μm for the PHR) and threaded to a silver probe and filled with conductive liquid (aCSF). The electrode was attached to a waterproof support with a side hole connected to a syringe, with which the nerve was suctioned into the electrode, forming an isolated medium inside it. Once in the electrode, the nerve will be connected to the silver probe due to the conductive liquid. Then, the matrix of organic electrodes was applied, surrounding the trunk of the left phrenic nerve. Signals were amplified and filtered (gain 0.5–10 K; BP 0.3–3 KHz), digitized (12 KHz, 16 bits; Tucker-Davis Technologies, Alachua, FL, USA) and exported using OpenEx software (Offline Sorter and NeuroExplorer; Plexon Inc., Dallas, TX, USA). Signal from the gold standard SE pENG will be noted in this work as $S^{SE}$ and those from each OE channel $C_{i,j}$ are named $S_{i,j}^{OE}$.

### 2.5. Data Processing

A data processing chain is proposed to compare phrenic bursts detection obtained from $S^{SE}$ (gold standard) with each $S_{i,j}^{OE}$. The proposed method (Figure 4) starts with noise reduction and envelope detection. Then it proceeds to the extraction of comparison indicators (correlation and ROC curves). Each step of the developed approach will be explained in the following subsections.

#### 2.5.1. Noise Reduction and Envelope Detection

All the acquired signals ($S_{i,j}^{OE}$ and $S^{SE}$) were bandpass filtered in the 300–3000 Hz interval using a 4th-order Butterworth filter [11]. When ECG artifacts are present (with an energy high enough to disturb phrenic bursts detection), morphological filters were applied on a moving 20-samples window, in order to reduce noise without disturbing the initial quality of the signal [37]. Then, a Savitzky–Golay filter was applied to smooth the signal without distortion and envelops were obtained through a Hilbert transform. Finally, envelopes were smoothed using a 20 Hz 4th-order Butterworth low-pass filter. The resulting signal envelopes ($E_{i,j}^{OE}$ and $E^{SE}$) were used to extract performance indicators (see Figure 4).

#### 2.5.2. Performance Indicators

In order to investigate the relations between OE and SE signals, two indicators were used: (1) correlation coefficient between $E_{i,j}^{OE}$ and $E^{SE}$, and (2) area under the curve (AUC) extracted from the receiver operating characteristic (ROC) curve produced to estimate the capacity of OE to detect spontaneous bursts using SE signal as reference.

Correlation coefficient: The Pearson correlation coefficient ($\rho_{i,j}$) was calculated to measure the degree to which $E_{i,j}^{OE}$ and $E^{SE}$ signals are linearly related:(1)$$\rho_{i,j}=\frac{cov(E_{i,j}^{OE},E^{SE})}{\sigma_{i,j}^{OE}\cdot \sigma^{SE}}$$
where cov is the covariance and $\sigma_{i}^{OE}$ and $\sigma^{SE}$ are, respectively, the standard deviation of $E_{i,j}^{OE}$ and $E^{SE}$ signals.

ROC curves and AUC: ROC curves were used to analyze the performance of a simple thresholding, applied on $E_{i,j}^{OE}$, to detect phrenic bursts. Reference classes were extracted from $E^{SE}$ by applying a threshold *K* fixed manually, by minimizing the number of visible phrenic bursts. Positive (P) and negative (N) classes were defined as sample sets belonging to a burst or not (Figure 5). Concerning the ROC curves constructed from the OE signals, 200 points were plotted for discriminating threshold values, λ, varying from 0 to M maximum value of the signal. We define prediction classes from those discriminating thresholds: predicted condition positive (True or T) for all samples greater than λ and predicted condition negative (False or F) for the lower ones. We define the outcomes as
TP=(P)∩(T);TN=(N)∩(F)
(2)FP=(N)∩(T);FN=(P)∩(F)

The ROC curve shows the true positive rate, or detection probability $P_{d}\left(\lambda \right)$, against false positive rate, or false alarm probability $P_{fa}\left(\lambda \right)$, for variable threshold and characterizes the detector of phrenic burst behaviour:$$P_{d}\left(\lambda \right)=\frac{\mathrm{TP}}{\mathrm{TP}+\mathrm{FN}}$$
(3)$$P_{fa}\left(\lambda \right)=\frac{\mathrm{FP}}{\mathrm{TN}+\mathrm{FP}}$$

Finally, the Area Under the Curve (AUC) is extracted from the ROC curve in order to evaluate the detection performance. The AUC value lies between 0.5 to 1, where 0.5 denotes a bad detector and 1 corresponds to an ideal detector. All analysis were performed using the commercially available software MatLab (Mathworks Inc., Natick, MI, USA).

Signal-to-noise ratio: For all the datasets, the SNR was calculated as the ratio between the mean absolute value (MAV) amplitude of the ENG signals recorded during bursts period and outside bursts period, based on P/N classifier defined on AUC and ROC curves part, as performed in [38]:$$\mathrm{MAV}=\frac{1}{N}\sum_{i=1}^{N}\left|x\right|$$
$$\mathrm{SNR}=20log_{10}\frac{mean(\mathrm{MAV}\left(ENG_{P}\right))}{mean(\mathrm{MAV}\left(ENG_{N}\right))}$$
where $ENG_{P}$ is ENG signal during bursts period and $ENG_{N}$ outside.

## 3. Results

This section presents: (1) results of one representative dataset, which was chosen because it includes a set of 16 OE channels, (2) global results from the whole dataset.

### 3.1. Example of Data Processing and Analysis

Figure 6 shows raw ($S^{SE}$, $S_{i,j}^{OE}$) and envelope ($E^{SE}$, $E_{i,j}^{OE}$) signals from one dataset. Phrenic bursts were clearly visible from the suction electrode (SE) signal. Their duration was ~0.5 s, and the respiratory rate was about 33 breath per minute, in accordance with previous results [34]. Most OE sites also displayed visible phrenic bursts. Table 1 describes the SNR for each recording site of the OE and the SE. As expected, the highest SNR was measured from the SE (11 dB before, and 15 dB after data processing). SNR from OE recording sites varied from 0.36 to 6.78 dB before processing and from 0.63 to 8.87 dB after processing. The mean SNR (±SD) after data processing was 5.51 ± 3.61 dB. Of note, ECG artifacts visible on some channels (see, for example, $S_{1,1}^{OE}$, $S_{1,3}^{OE}$, and $S_{4,2}^{OE}$ in Figure 6) were removed after data processing. Correlation coefficients between all signals were calculated from the whole dataset. Figure 7A shows the correlation map of the example dataset. The last row represents the correlation coefficients of every OE channel with SE which values are indicated in Table 1. The mean correlation coefficient (±SD) after data processing was 0.83 ± 0.19 (range 0.39 to 0.93). However, 3 out of 16 channels had low correlation coefficient values (<0.6). Figure 7B presents the detection probability of phrenic bursts recorded by the 16 OE sites (ROC curves obtained by comparing OE signals with the SE signal). The mean PHR burst AUC (±SD) after data processing was 0.94 ± 0.07 (range 0.77 to 0.99) (Table 1). Twelve out of 16 channels had an AUC greater than 0.9.

### 3.2. Global Results

The measured EIS calculated from 10 electrodes was 31,668 ± 1866 Ω (mean ± SD) at 1 kHz. Table 2 represents averaged results obtained from 9 rats. Correlations ρ(i,j) ranged between 0.047 and 0.997, and average value is 0.736 ± 0.135. Minimum and maximum values of AUC were 0.63 and 0.99, respectively. Mean AUC values (± SD) were 0.900 ± 0.091. In each dataset, at least one channel showed a correlation coefficient higher than 0.7, and at least one AUC was above 0.9.

## 4. Discussion

The objective of this study was to investigate the performances of a self-locking custom-designed multicontact organic electrode (OE) technology to record from a small nerve after acute implantation. To this aim, we recorded from the phrenic (PHR) nerve in the rat working heart–brainstem preparation, which preserves physiological cardiorespiratory functions [34,39,40]. A dual PHR ENG recording protocol was achieved by applying a suction electrode (SE) at the cut-end of the PHR nerve to obtain a high quality signal, and wrapping the OE around the PHR epineurium. We performed data filtering to improve PHR ENG burst detection on our signals and analyzed the quality of the OE recordings compared with SE signals. Although we do not provide a comparison with respect to classical metallic electrode technologies, our results show that good quality recordings could be obtained with small PEDOT:PSS recording sites at low impedance. Furthermore, we were able to bring out clear inspiratory activity from signals that were prima facie expressing no respiratory activity. These results suggest that OEs are suitable to record respiratory signals and show the feasibility to easily install 16 recording sites around a small nerve, therefore improving spatial resolution.

### 4.1. Electrophysiological Relevance of PEDOT:PSS

One of the main advantages to use the organic conducting polymer PEDOT:PSS is its mixed ionic/electronic conductivity making it a good candidate for bioelectronics applications in general. The ability of the material to absorb water and thus to uptake dissolved ions, making the entire bulk of the material participating in the interaction between ions and electrons, makes it a good candidate for bioelectronics applications in general. The electrical aspect of it results in a difference in the capacitive component of the electrode: for metal electrodes, the capacitance is surfacic, while for PEDOT:PSS, the capacitance is volumetric [41]. This means that for the same electrode surface, the use of PEDOT:PSS results in an increase in the capacitance. As a consequence, considering a fixed electrode surface, the use of PEDOT:PSS allows to achieve a lower impedance, thus a better signal to noise ratio for recording purposes compared to metal electrodes [42]. Furthermore, planar metallic electrodes such as IrOx based devices might increase unwanted faradaic contributions, tolerable only if redox processes are confined and reversible [43]. Venkatraman et al. also compared PEDOT:PSS electrodes with PtIr and IrOx electrodes, and found out that PEDOT:PSS coated microelectrodes exhibit superior performances than PtIr electrodes in terms of both recordings and stimulation and present as well superior charge injection limits than IrOx electrodes at 0 DC bias voltage [44].

### 4.2. Suction Electrode for ENG Recording

SE are commonly used to record ENG from transected axons of various nerves in situ [34,40,45,46]. They provide very high-quality ENG signals that can be used to monitor the central breathing activity, and also to compute spike–trigger averaging in electrophysiological studies [47,48]. The quality of our SE signals is thus in agreement with previous observations. This is likely due to the fact that a small part of the distal nerve, including the cut-end, is aspirated and inserted within the SE tip which contains a small amount of conducting liquid. Consequently, action potentials propagating along the nerve reach the distal nerve portion almost simultaneously. Thus, it produces a high current density within the SE, and this results in high-amplitude compound action potentials. Such ENG signals recorded with SE is classically used as the gold standard for nerve recordings. However, for obvious reasons, nerve transection cannot be considered in humans. A challenge thus remains to design appropriate electrodes that can be applied at the surface of the nerve while providing useful ENG signals.

### 4.3. Acute ENG Recordings with OE

Different types of self-locking electrodes are available, and some of them are used in humans for neuromodulation [49]. Here, we provide evidence for good PHR ENG acute recordings using PEDOT:PSS electrode that include a self-locking part. When locked, the electrode diameter is 350 μm (see Figure 2). The rat phrenic nerve varies from 250 to 320 μm in diameter [50]. Thus, the internal cuff diameter of our OE is larger than the external diameter of the PHR nerve and should limit nerve damage. OEs are made with a thin biocompatible polymer (parylene) and use conducting polymers called PEDOT:PSS coupled with gold wires, conferring a low impedance and flexibility. Recordings performed on this paper are acute ones. Even if our electrodes have high flexibility, chronic experiments can lead to drastically different results such as the materials, but also the stiffness and various form factors influencing the outcome. This technology has been shown to provide high signal-to-noise ratio recordings of the neural interface on electroencephalographic and electrocorticographic applications [24,35,51,52,53,54]. OEs have also been used to record neuronal activity in patients in the operating room [55]. Moreover, OE sites can be used to both record and stimulate neurons [52]. Compared to the $S^{SE}$, the $S^{OE}$ had a lower signal-to-noise ratio. This likely resulted from the transfer function and signal attenuation from the signal sources within the nerve to the recording sites on the epineurium. A limitation of this study was the lack of impedance measurement of the various OE sites when the electrode was in contact with the nerve. Higher impedance on some OE sites may be responsible for a decrease in pENG amplitude and AUC, and thus weak correlations between $S^{OE}$ and $S^{SE}$. However, our results demonstrated that both the $S^{OE}$ and $S^{SE}$ had a similar pattern. This pattern included a step increase in discharge at the beginning of inspiration, a ramp-like increase throughout the inspiratory phase, and a sharp decrease in PHR activity at the end of inspiration, reflecting a physiological PHR discharge [34,39,40,45,46,56]. Furthermore, high correlation coefficients were measured between $S^{SE}$ and $S^{OE}$. In addition, elevated AUC scores were found on several OE channels in each preparation. Our results suggest that OE may represent an alternative to more classical electrodes that are currently used for neuromodulation applications. Recordings of the PHR nerve activity have been made in cats using simple metallic cuff electrode, and the averaged signal to noise ratio (SNR) was 2.44±0.18 dB [17]. Here, the technology proposed in this paper allow us to record from the rat PHR nerve with a higher averaged SNR (5.51±3.61 dB). Helical, cuffs or flat electrodes, with mono-, bi-, or tri-polar characteristics have been mainly tested for neurostimulation [57,58,59]. Further investigations are needed to compare the performances of those electrodes with multicontact OE in ENG recording. Furthermore, our study was performed in acute condition, while neuromodulation studies require in vivo chronic implantation. Although the stability of PEDOT:PSS-coated electrodes has been previously reported, their reliability might be improved before it can be used in chronic condition [60]. One of the main issues in terms of long-term stability of PEDOT:PSS-coated electrodes remains the delamination of such conducting polymer from the electrode surface. This comes from the poor adhesion properties of such materials on the top of metal layers in aqueous conditions, even more pronounced under stimulation. Future approaches may overcome those technological limitations, as for instance using biostable adhesion promoters [61]. Fibrosis was not investigated in our study although this problem (as well as other nerve damage) has been reported with commonly used metallic electrodes. As evidences for tissue damage were seen 1 day post-implantation [62], we hypothesize that fibrosis did not develop in our acute conditions. A recent study using similar Parylen C biosensors suggested a very good biostability after 6 months of brain recording activity [63]. Further investigations after chronic nerve implantation with PEDOT:PSS electrode are needed to determine if fibrosis occurs.

### 4.4. Towards a Better Selectivity to Record and Stimulate with OE?

In this study, we designed an OE with symmetrical sites adapted for ENG acquisition. We succeeded to use this OE as a cuff electrode with most of the OE sites in contact with the nerve, allowing multiple pENG recordings. Such a recording technique may be transposed to other peripheral nerves such as the vagus nerve which is the target of neuromodulation in a number of pathologies [64,65,66]. As stimulation of the vagus nerve with an OE has been shown to elicit differential cardiac responses, future studies should test the possibility to both stimulate and record with the same multicontact OE, probably using different contacts.

The phrenic nerve is mainly composed of motor fibers, and is also composed of 30–45% sensory fibers [67]. The vast majority of motor fibers (97%) correspond to alpha-axons, and the rest (3%) are gamma-axons [68]. As this nerve was cut, signals recorded in pENG corresponded only to active motor fibers. Our results did not provide evidence for differentiated compound action potentials between the various OE sites channels. This apparent lack of spatial selectivity may be explained by several factors. The very small diameter of the PHR likely reduces transversal (or fascicular) selectivity [69]. Furthermore, the fast conduction velocity of PHR motor fibers (~55–60 m·s${}^{-1}$ [70]) associated with the rather short distance between OE sites may limit the velocity selectivity [71]. It is therefore possible that similar multiple ENG recordings with OE applied to bigger peripheral nerves or designed with longer distances between OE sites may be able to achieve fascicular and/or velocity selectivity, and thus differentiate distinct pools of nerve fibers. Moreover, the signal processing methods can be improved to better estimate complementary information from the different acquired sites. Achieving successful spatial selectivity of phrenic nerve activity could foster researches on axonal transmission pathologies or unexplained respiratory failures.

## 5. Conclusions

The main contribution of this study was to show the ability of custom biocompatible OE to record respiratory-related nerve signals and to detect physiological bursts on PHR activity from a matrix of electrodes placed around the phrenic epineurium. We also provided evidences for an optimization of respiratory-related burst detection after signal processing, and the measurements of correlations between OE channels and SE envelopes as well as ROC curves demonstrated a good performance of the OE matrix. OE appear as a promising tool to record from multiple ENG signals in peripheral nerves. Future works will be directed towards the improvement of the OE geometry and the related signal processing methods in order to improve selective recording, as well as the application and evaluation of this OE technology for estimating useful ENG-derived control variables for closed-loop neuromodulation therapy.

## Figures and Tables

**Figure 1 sensors-21-05594-f001:**
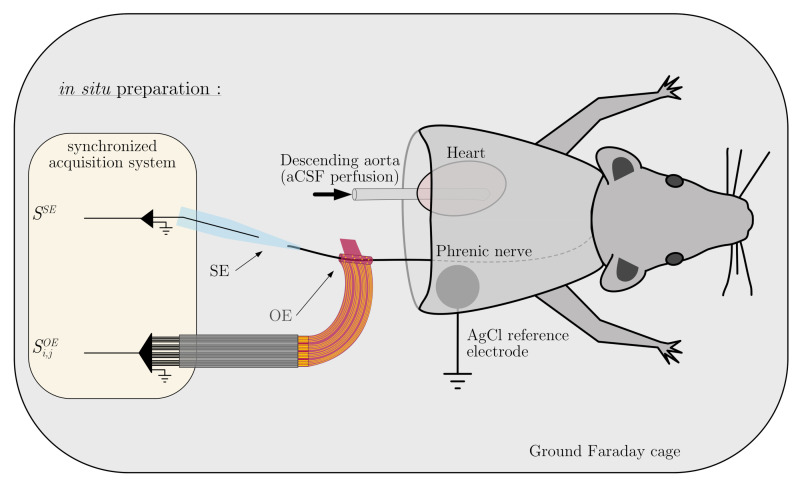
Schematic representation of the in situ preparation. Perfusion was applied through the descending aorta using a warmed (31 ${}^{\circ }$C) artificial cerebrospinal fluid (aCSF) equilibrated in carbogen. A pump with an adjustable flowrate was used to control perfusion pressure, and ensure an adequate oxygenation of the brainstem. Cardiorespiratory functions resume 5–10 min after re-perfusion, and the preparation generates an eupneic pattern of breathing. Serial recordings of the phrenic nerve discharge were made with SE attached to the cut end, and OE attached to the main nerve trunk. All signals were acquired synchronously with the same acquisition system.

**Figure 2 sensors-21-05594-f002:**
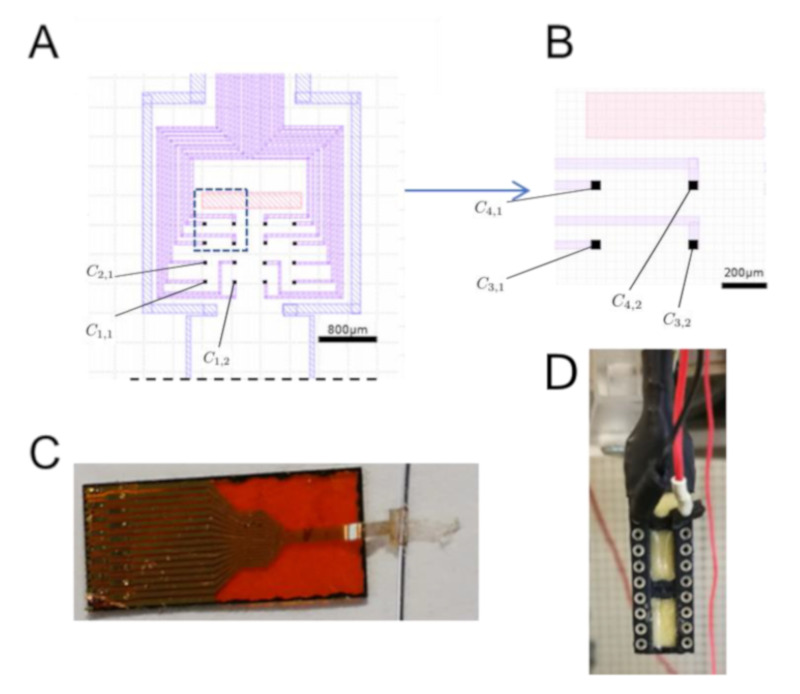
OE representation. (**A**) Schematic view of the 16 (4 × 4) recording sites with in purple gold wires; blue PaC boundaries; red gap to insert the tongue (upper panel) and dark purple the recording sites. The PaC tongue of the organic electrode is not fully represented here (dotted line) and is 4.4 mm long. (**B**) Zoomed view of the blue doted square in A representing 4 gold/PEDOT:PSS recording sites 20 × 20 μm. A matrix notation of the form $C_{i,j}$ is used here to identify a given individual channel in row *i* and column *j*. The nerve lies along the horizontal axis on both top figures. (**C**) Whole picture of the organic electrode mounted on a kapton layer, with the recording sites wrapped around a 350 *∅*μm wire. (**D**) The reference is an Ag/Cl electrode connected to the black wire disposed under the ribcage of the animal and the red wire (the ground) is connected to a Faraday cage.

**Figure 3 sensors-21-05594-f003:**
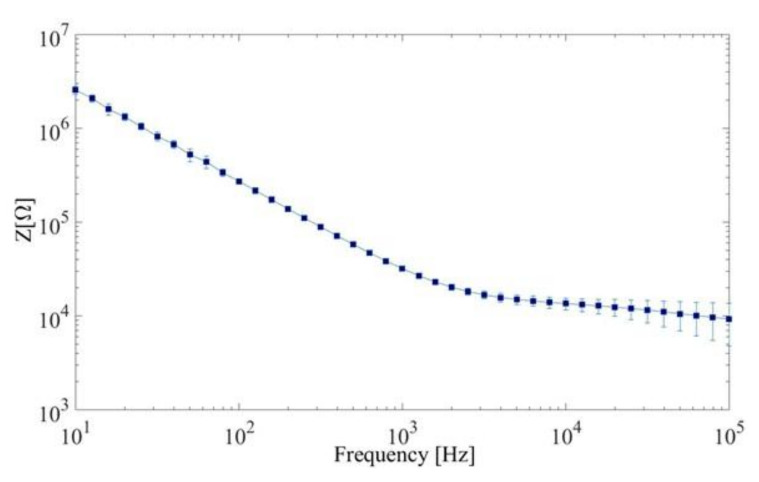
Electrochemical impedance spectrum of PEDOT:PSS-coated electrodes. Ten electrodes were used for these measurements.

**Figure 4 sensors-21-05594-f004:**
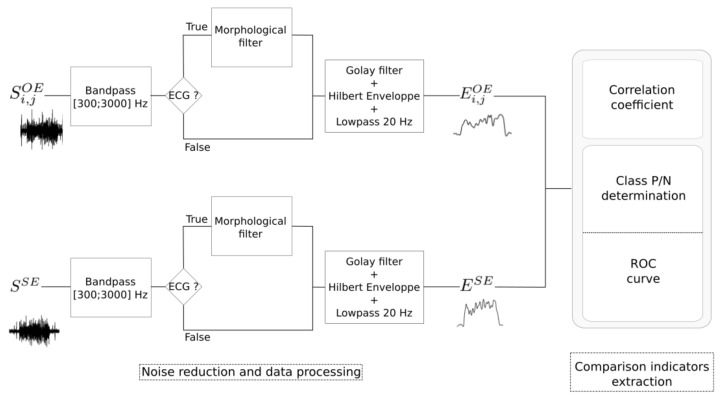
Description of data processing and feature extraction applied to OE and SE signals. All available channels from the OE are processed. Every channel is compared with the filtered SE signal, used as gold standard, in order to estimate performance.

**Figure 5 sensors-21-05594-f005:**
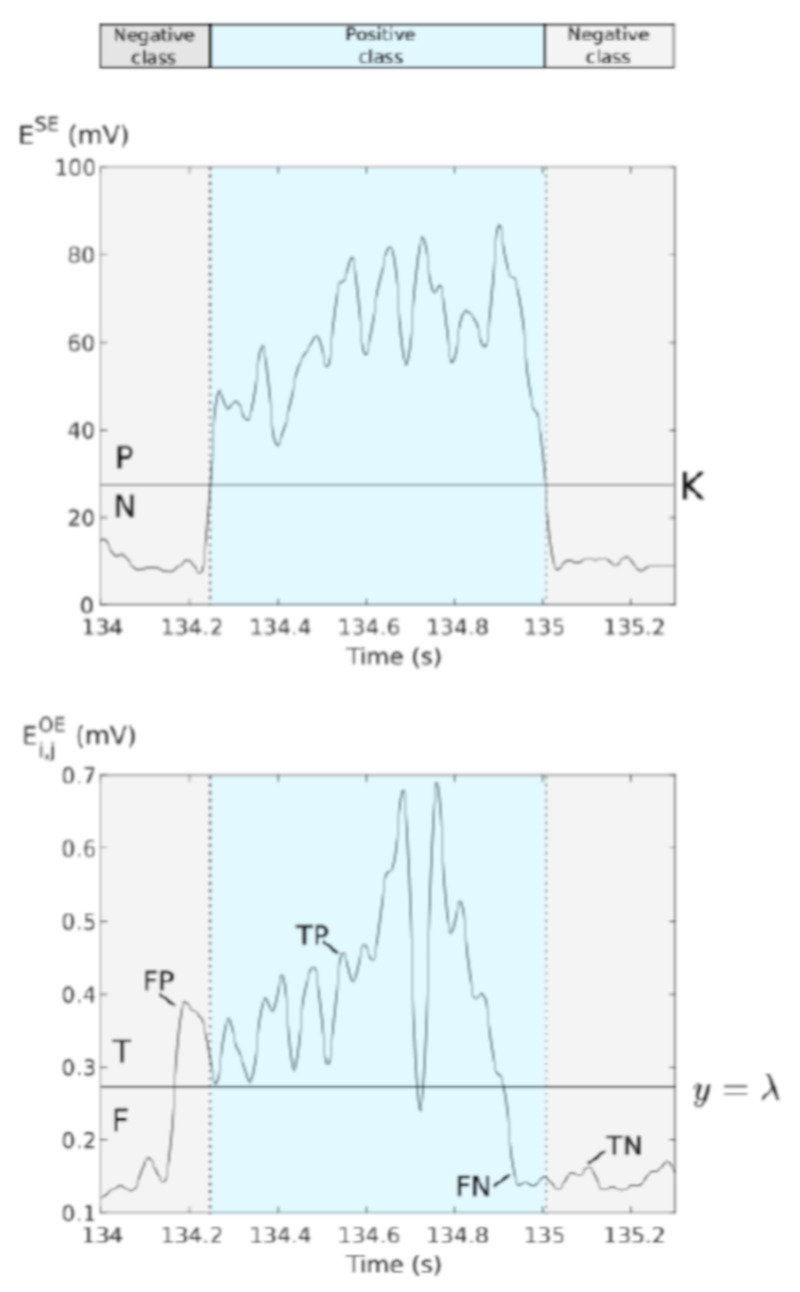
Classifier determination. SE signal is used as reference to create our classifier. Positive class (P) is defined as “all points above the threshold” and negative class (N) the opposite. The same definition is applied to the OE signal for “True”-class (T) and “False”-class (F) with the threshold defined as y=λ. We define True positive (TP) all samples belonging to (P) and (T), True Negative (TN) those belonging to (N) and (F), False Positive (FP) to (N) and (T) and False Negative (FN) to (P) and (F).

**Figure 6 sensors-21-05594-f006:**
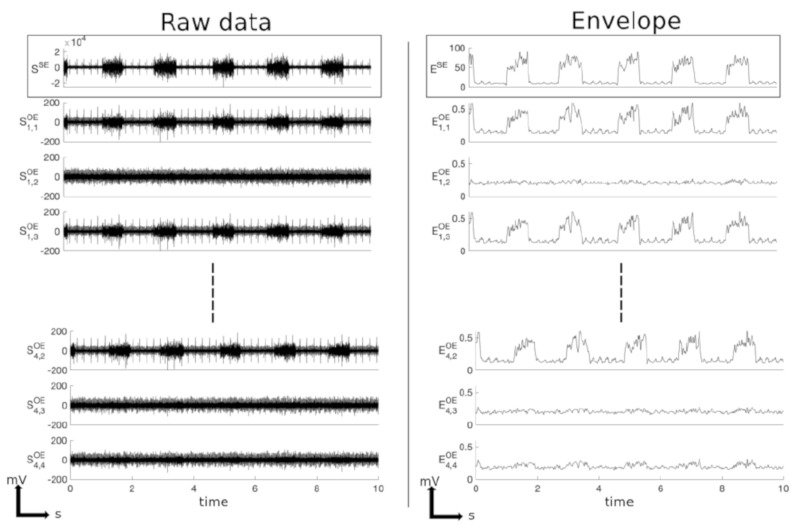
**Left**: Raw data from one representative example of the acquired dataset. **Right**: Output of the data processing step, showing the envelope of a denoised signal. **Top panel**: Phrenic signal provided by gold standard suction electrode with typical phrenic waves and electrocardiogram artifacts and Phrenic integrated signal provided by gold standard suction electrode. **Bottom panels**: Phrenic signal obtained from the OE multipolar cuff electrode. This example underlines the heterogeneous properties of these different recordings and the kinds of noise typically observed in this setup.

**Figure 7 sensors-21-05594-f007:**
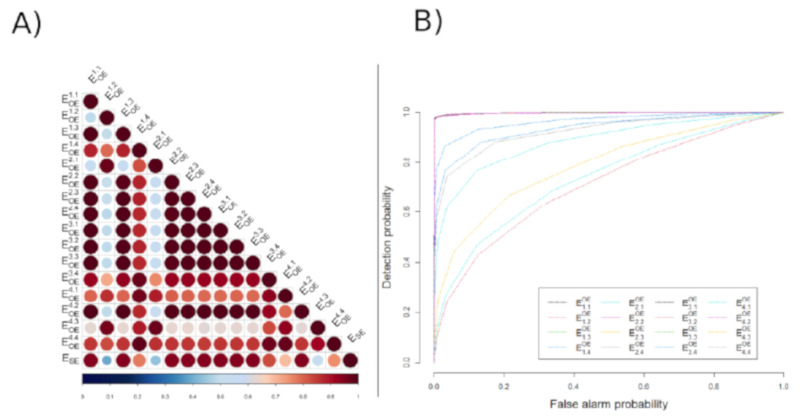
(**A**) Correlation map between signal provided by gold standard SE and OE. (**B**) Receiver operating characteristic curve illustrating the detection of phrenic waves based on constant thresholding on gold standard signal. Those scores reflects the correlation scores. The lower the correlation is, the lower the AUC is.

**Table 1 sensors-21-05594-t001:** Results of the proposed phrenic waves detection of the dataset after filtering and ECG removal steps. Second and sixth line are correlation coefficient between OE signal and SE signal for every channel, third and seventh line AUC results using SE signal as true detect, and fourth and eighth lines are SNR in dB.

Channels	1,1	1,2	1,3	1,4	2,1	2,2	2,3	2,4
Correlation	0.934	0.391	0.933	0.768	0.439	0.927	0.932	0.933
AUC	0.994	0.708	0.994	0.929	0.734	0.992	0.994	0.993
SNR (dB)	8.705	0.678	8.574	2.278	0.745	7.188	8.396	8.758
3,1	3,2	3,3	3,4	4,1	4,2	4,3	4,4	Mean ± STD
0.934	0.934	0.933	0.821	0.686	0.935	0.523	0.748	0.798 ± 0.191
0.994	0.994	0.994	0.958	0.879	0.994	0.783	0.918	0.925 ± 0.072
8.811	8.701	8.861	2.974	1.611	8.874	0.980	2.063	5.510 ± 3.61

**Table 2 sensors-21-05594-t002:** Global performance of the proposed phrenic waves detection. Column 2 indicates the number of channel used for recording; Columns 3 to 5 and columns 6 to 8 show, respectively, correlation performance and AUC performance. Last line shows the global mean of each marker. On recordings 9 and 10, only 1 channel was correctly acquired, min and max interval and standard deviations are thus not provided.

Rat	Chan.	Corr. Mean	Corr. Interval	Corr. std	AUC Mean	AUC Interval	AUC std
1	16	0.798	[0.391;0.935]	0.191	0.925	[0.708;0.994]	0.072
2	3	0.872	[0.858;0.879]	0.012	0.998	[0.998;0.999]	$5.8\times 10^{-4}$
3	8	0.661	[0.047;0.997]	0.175	0.860	[0.557;0.967]	0.3553
4	8	0.398	[0.097;0.960]	0.275	0.816	[0.628;0.999]	0.1207
5	8	0.656	[0.166;0.940]	0.297	0.908	[0.648;0.999]	0.1600
6	8	0.549	[0.451;0.945]	0.169	0.825	[0.758;0.999]	0.0769
7	16	0.582	[0.505;0.702]	0.052	0.867	[0.819;0.943]	0.0337
8	8	0.954	[0.915;0.995]	0.042	0.999	[0.996;1]	$1.75\times 10^{-4}$
9	3	0.972	[0.969;0.973]	0.003	0.999	[0.998;0.998]	$8.55\times 10^{-5}$
Mean		0.736	[0.610;0.923]	0.135	0.900	[0.852;0.989]	0.091

## Data Availability

The data that support the findings of this study are available from the corresponding author, V.L.R., upon reasonable request.

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
