# Peer review of "Assessment of the Use of Multi-Channel Organic Electrodes to Record ENG on Small Nerves: Application to Phrenic Nerve Burst Detection"

_sensors, 2021, doi:10.3390/s21165594_

Round 1

Reviewer 1 Report

The paper deals with the design and preliminary test of a multicontact neural electrode. The experimental work is particularly carried out with care but I have serious concerns about the main claim of the paper: the novelty and the rationale of the design. Indeed the authors claim at the end of the paper that "to our knowledge it is the first time...". It is maybe the case considering the technology AND the phrenic nerve but it does not induce a scientific novelty. To my knowledge, both the technology and phrenic nerve ENG have been achieved in previous studies. I would like to hemp authors to enhance their paper by clarifying their main contribution based on a scientific hypothesis that they can assess and compare to the state-of-the-art.

Detailed major comments only:

INTRODUCTION: ENG recording through multicontact cuff electrodes is a huge area of research. PEDOT:PSS and Parylene C as soft support were already used for such a type of electrode. That's however true that PEDOT:PSS is less studied than PEDOT alone but recent papers were issued and not cited in the present one (i.e. 10.1038/s41598-019-46967-2 but hey are few others) and at large mutlicontact nerve recording is plenty of papers based on PEDOT PtIr but also IrOx known to have very good properties. On the phrenic nerve Sahin published a paper on ENG recording with a simple cuff (10.1152/jappl.1997.83.1.317). As a whole the authors should detail the the nerve cuff recording at large and then explain what they are proposing as a novel approach. On one hand, the Shain paper seems to show that a classical cuff is enough for Phrenic Nerve ENG so why using a multicontact PEDOT:PSS technology for this nerve? And for other nerves for which ENG is more challenging (for instance siactic nerve) the paper should have performed the corresponding in vivo trial.

METHODS: 2.1 is clearly described. 2.2 is lacking of many information. If authors want to promote PEDOT:PSS they should go through much more assessments of the technology. For instance EIS measures (impedance @1kHz is  ot enough) should be given.

The rationale for the grid dimension as it is known that for ENG recordings longitudinal spacing is of prime importance and linked to the wavelength. Even though monopolar recordings are performed these distances are still meaningful in a time-space view of ENG propagation.

Nothing is said about the type of fibres supposed to be recorded (it influences these dimensions).

The schematic is incomplete, is the animal connected to the reference? And how? units seem to be incorrectly printed in the pdf file.

Threshold setting in particular Lambda is not explained.

Concerning the method, as authors claim their approach is better than existing cuff electrodes, the comparison with suction electrode is to questionable. It would be better to still use SE as gold standard but use a classical tripolar cuff to investigate the benefit of such a technology.

RESULTS: Avoid to use qualitative statements when an objective measure exists: "Noise is lower" should be replaced by SNR in all channels +SE.

There is no statistics on the 13 rats to check overall significance.

Impedance, in particular for recording, is very important to know. Did you measure the "in vivo" impedance that may explain also the discrepancy between contact. Without this measure you may have a confusion factor.

DISCUSSION: the first limitation of the study to be stated is that this is acute recordings. All cuff designs may show drastically different results in a chronic experiments as not only the material but also the stiffness and various form factors influence the outcome.

You state that you obtained "high quality signals" but how do you quantitatively assess this assertion? At least again SNR should be computed.

What is the interest of a multicontact cuff? It is not discussed and I think you cannot show, with the phrenic nerve, the interest of such a technology if a classical cuff can record safely and robustly the Phrenic nerve burst.

Finally, you show that PEDOT:PSS can be used to record ENG but again authors fai to prove that this recording is better than IrOx contacts for instance or at least PtIr contacts.

As a whole the paper lacks of too much objective quantification and a robust comparison with the state-of-the-art. I hope I give some hints for authors to deeply revised the proposed research.

Reviewer 2 Report

Authors presented flexible multi-channel organic electrodes and its capability of sensing nerve bursts from phrenic respiratory nerve electroneurograms. The experiments are well established, and results are interesting. I recommend the journal after following revision.

* It is difficult to see the novelty of organic electrodes. PEDOT:PSS electrode and its electrical characteristic on the flexible substrate have been published in other journals. I would suggest to focus on the novelty of application using flexible electrode rather than the structure and fabrication of organic electrodes in the main text.

* It is difficult to compare the performance of electrodes with other neural interface as there are not much information to be compared. For example, the author claimed that the electrodes have advantages due to their porous nature. Yet, the text does not include the improvement (such as impedance or charge capacity) or that advantage compare to other electrodes.

* The common obstacle with PEDOT:PSS electrode is the reliability and its chronic usage in the body. Can author please confirm any of these in bench top test?

Round 2

Reviewer 1 Report

The paper has been deeply edited and enhanced. I'm still not fully convinced about the novelty but as it comes from the design of the study (choice of the targeted nerve that does not include complex ENG and no comparison with standard cuff technology) it cannot be changed further.

The paper is however well written and the experiment rigorously conducted. It is thus an incremental research that adds a bit of knowledge about OE electrodes, but not so much. I fear that a new round of review won't drastically change the content but if the paper is considered for publication I suggest the following further considerations:

  • comment 1: OK as a whole but few remaining remarks. First even though ref 17 has been added and shortly described in the introduction, a short comparison, as it is an alternative way of measuring Phrenic nerve ENG, should be discussed in the discussion section.
  • comment 1: "We optimized PHR ENG bursts detection" I do not see what you optimize.
  • comment 1: you claim to assess a self-locked electrode adapted to small nerve but this kind of technology already exists even for small multi contact electrodes (https://www.cortec-neuro.com/product-portfolio/airray-research-helix-cuff/) and furthermore the current paper does not show this property (there is no data about electrode to nerve fitting)
  • comment 2: OK
  • comment 3: OK but the EIS is not discussed. It shows a capacitive like electrode in series with a resistor. A short comparison with EIS of classical metal contact is needed to support your claim about the interface advantage (be careful to be area independent).
  • comment 4: OK but about selectivity you have 2 different things you should not mismatch: the transversal so called fascicular selectivity and you are right it is linked to the ratio of contact's number and nerve diameter, and the velocity selectivity i.e. fibres’ type selectivity linked to the longitudinal distance between contacts (see Donaldson papers on this topic). Not clear in your explanation.
  • comment 5: OK
  • comment 6: OK
  • comment 7: as stated in my introduction cannot be changed. So OK except add, as mentioned before, a short comparison with [17] in the discussion.
  • comment 8: OK
  • comment 9: missing
  • comment 10: OK
  • comment 11: OK but you have also the problem of fibrosis. It is directly linked to the type of material and so of strong impact: to be studied in the case of a material assessment.
  • comment 12: your answer further confirms that you cannot show the interest in this particular study. You then have to focus on the interest of the PEDOT:PSS material use.
  • comment 13: OK
  • comment 14 was a global remark: OK

Considering all my remarks I think the title should be changed to something linked more directly to OE electrodes such as (but it is up to you) "assessment of the use of multi-channel OE electrodes to record ENG on small nerves: application to phrenic nerve burst detection"

Reviewer 2 Report

I have no more comment. 
